# Arginase Inhibition Mitigates Bortezomib-Exacerbated Cardiotoxicity in Multiple Myeloma

**DOI:** 10.3390/cancers15072191

**Published:** 2023-04-06

**Authors:** Aleksandra Paterek, Marta Oknińska, Zofia Pilch, Anna Sosnowska, Kavita Ramji, Urszula Mackiewicz, Jakub Golab, Dominika Nowis, Michał Mączewski

**Affiliations:** 1Department of Clinical Physiology, Centre of Postgraduate Medical Education, 99/103 Marymoncka Street, 01-813 Warsaw, Poland; aleksandra.paterek@cmkp.edu.pl (A.P.); marta.okninska@cmkp.edu.pl (M.O.); urszula.mackiewicz@cmkp.edu.pl (U.M.); 2Department of Immunology, Medical University of Warsaw, 5 Nielubowicza Street, 02-097 Warsaw, Poland; zofia.pilch@wum.edu.pl (Z.P.); anna.alicja.sosnowska@gmail.com (A.S.); kaveeta@gmail.com (K.R.); jakub.golab@wum.edu.pl (J.G.); dominika.nowis@wum.edu.pl (D.N.); 3Centre of Preclinical Research, Medical University of Warsaw, 1B Banacha Street, 02-097 Warsaw, Poland; 4Laboratory of Experimental Medicine, Medical University of Warsaw, 5 Nielubowicza Street, 02-097 Warsaw, Poland

**Keywords:** multiple myeloma, cardiotoxicity, bortezomib, arginase inhibitor, nitric oxide, endothelial function

## Abstract

**Simple Summary:**

Multiple myeloma is a hematological malignancy associated with increased cardiovascular morbidity and mortality. Its therapies also result in adverse cardiac effects. Since multiple myeloma is associated with increased expression of arginase, an enzyme that consumes an amino acid arginine, a precursor for nitric oxide synthesis, our aim was to test if cardiotoxicity mediated by multiple myeloma and one of its most commonly used therapies, bortezomib (a proteasome inhibitor), can be ameliorated by an arginase inhibitor through improved vascular nitric oxide system. We used a mouse Vĸ*MYC model of multiple myeloma. The disease resulted in progressive cardiac dysfunction, and bortezomib exacerbated this effect. An arginase inhibitor protected the heart against bortezomib- or multiple myeloma-induced toxicity but did not completely prevent the effects of the multiple myeloma+bortezomib combination. Neither multiple myeloma nor bortezomib impaired vascular nitric oxide system. Our study suggests that proteasome inhibitors should be used with caution in patients with advanced myeloma, where the summation of cardiotoxicity could be expected. Therapies aimed at the nitric oxide pathway, in particular arginase inhibitors, could offer promise in the prevention/treatment of cardiotoxicity in multiple meyloma.

**Abstract:**

Background: Multiple myeloma (MM) is associated with increased cardiovascular morbidity and mortality, while MM therapies also result in adverse cardiac effects. Endothelial dysfunction and impaired nitric oxide (NO) pathway is their possible mediator. Objective: Since MM is associated with increased arginase expression, resulting in the consumption of ʟ-arginine, precursor for NO synthesis, our aim was to test if cardiotoxicity mediated by MM and MM therapeutic, bortezomib (a proteasome inhibitor), can be ameliorated by an arginase inhibitor through improved endothelial function. Methods: We used a mouse Vĸ*MYC model of non-light chain MM. Cardiac function was assessed by echocardiography. Results: MM resulted in progressive left ventricular (LV) systolic dysfunction, and bortezomib exacerbated this effect, leading to significant impairment of LV performance. An arginase inhibitor, OAT-1746, protected the heart against bortezomib- or MM-induced toxicity but did not completely prevent the effects of the MM+bortezomib combination. MM was associated with improved endothelial function (assessed as NO production) vs. healthy controls, while bortezomib did not affect it. OAT-1746 improved endothelial function only in healthy mice. NO plasma concentration was increased by OAT-1746 but was not affected by MM or bortezomib. Conclusions: Bortezomib exacerbates MM-mediated LV systolic dysfunction in a mouse model of MM, while an arginase inhibitor partially prevents it. Endothelium does not mediate either these adverse or beneficial effects. This suggests that proteasome inhibitors should be used with caution in patients with advanced myeloma, where the summation of cardiotoxicity could be expected. Therapies aimed at the NO pathway, in particular arginase inhibitors, could offer promise in the prevention/treatment of cardiotoxicity in MM.

## 1. Introduction

Multiple myeloma (MM) is the second most common hematological malignancy in adults, characterized by clonal proliferation of malignant plasma cells in the bone marrow, monoclonal protein in the blood and urine, and end-organ damage [1]; the latter differentiates active MM from monoclonal gammopathy of undetermined significance and smoldering multiple myeloma. Infection and renal failure are the main causes of early mortality. Despite significant progress in therapy, MM remains an incurable disease with a 5-year relative survival rate of 53.9% [2].

MM is associated with increased cardiovascular morbidity and mortality: the standardized cardiovascular mortality ratio among MM patients is 1.84 times higher than in the general population [3]; however, the interpretation of these data is difficult since MM is a highly variable disease associated with multiple genetic aberrations and clinical subtypes, each potentially affecting the cardiovascular system in a different way. Moreover, many MM treatments exert their own cardiotoxic effects. Recently monoclonal gammopathy of undetermined significance has also been shown to be associated with increased cardiovascular morbidity [4].

Light chain MM can be associated with cardiac amyloidosis [5], resulting in restrictive cardiomyopathy and heart failure with preserved ejection fraction [6] and poor prognosis, although it has also been linked with systolic dysfunction [7]. Non-light chain MM is also associated with cardiac toxicity; its mechanism is even less clear. Hyperviscosity related to increased serum concentrations of monoclonal protein, expanded plasma volume, and reduced coronary reserve may cause systolic dysfunction and heart failure [8]. High-output heart failure, potentially caused by arteriovenous shunts, enhanced vascularity, hyperviscosity, and increased shear stress with increased NO production, has also been reported in MM patients [9].

Experimental studies indicate that increased plasma viscosity results in increased NO synthesis [10] through stimulation of endothelial nitric oxide (NO) synthase [11]. On the other hand, we have shown that in the mouse model of non-light chain MM, systemic concentrations of ʟ-arginine, a precursor for NO synthesis, are progressively reduced, correlating with arginase expression in myeloid cells [12]. Therefore, the effects of MM on endothelial function and systemic NO availability remain unknown.

Currently, MM therapy is based on combination regimens, and proteasome inhibitors (PIs) are their essential component. They include bortezomib, carfilzomib, and ixazomib [13]. PIs target proteasome, an enzymatic complex responsible for the degradation of misfolded or damaged proteins in the cell [13], causing the accumulation of those proteins in the endoplasmic reticulum. MM cells are particularly susceptible to proteasome inhibition due to their high immunoglobulins production rate. Since proteasome also degrades mediators of cell-cycle progression, apoptosis, angiogenesis, and inflammation, PIs potentially also affect these cellular functions [14].

Cardiovascular adverse events are common during PI treatment and were reported to occur in 51% of patients treated with carfilzomib and 17% of those treated with bortezomib [15] and include heart failure, hypertension, and thromboembolism. The exact mechanisms are unknown. We have previously shown [16] that bortezomib exerts reversible reduction of left ventricular ejection fraction (LVEF) in healthy rats and mice, probably related to abnormalities of mitochondrial respiration [16]. Furthermore, proteasomal inhibition leads to the downregulation of endothelial NO synthase activity, resulting in decreased NO availability and the potential development of hypertension and heart failure [17].

Thus the aims of the study were to (1) analyze the cardiac effects of untreated MM in the mouse Vĸ*MYC model of non-light chain MM recapitulating all aspects of human disease [18], including IgG overproduction, (2) test the hypothesis that bortezomib exacerbates cardiotoxicity related to MM itself, (3) test the hypothesis that bortezomib induces endothelial dysfunction and NO deficiency and that an arginase inhibitor, by supplying ʟ-arginine, prevents endothelial dysfunction from alleviating bortezomib-induced cardiotoxicity.

## 2. Materials and Methods

### 2.1. Mice

All experiments were performed in 8–12-week-old female C57BL/6 mice obtained from the Animal House of the Medical Research Center, Polish Academy of Sciences (Warsaw, Poland). Animals were housed in controlled conventional environmental conditions animal facility of the Medical University of Warsaw and Medical Centre for Postgraduate Education in Warsaw, with water and food provided ad libitum. The experiments were performed in accordance with the guidelines approved by the 1st Local Ethics Committee in Warsaw (approval No. 618/2018) and in accordance with the requirements of the EU (Directive 2010/63/EU) and Polish (Dz. U. poz. 266/15.01.2015) legislation.

### 2.2. Vκ*MYC MM Model

C57BL/6 mice were intravenously transplanted with 1 × 10^6^ Vĸ*MYC cells (a kind gift from Prof. Leif Bergsagel, Mayo Clinic College of Medicine, USA). In this syngeneic MM model, the disease develops in the spleen and in the bone marrow. Vĸ*MYC MM model tightly recapitulates human MM features, including bone and kidney involvement [18] as well as anti-myeloma drug sensitivity [19]. From week 2. until week 6. MM development was monitored with serum protein electrophoresis (SPEP) using Sebia Hydragels and HYDRASYS analyzer. The relative density of the bands corresponding to monoclonal protein was measured using ImageJ ver. 152a (NIH, Bethesda, MD, USA).

### 2.3. Treatments

Bortezomib was purchased from Adamed, Poland, dissolved in 0.9% NaCl and administered at 0.5 mg/kg intraperitoneally (i.p.). Arginase inhibitor OAT-1746 (OATD-02), a potent dual intracellular arginase 1/2 inhibitor entering clinical trials in cancer immunotherapy [20], was provided by Molecure SA (previously OncoArendi Therapeutics SA), Warsaw, Poland. OAT-1746 was dissolved in PBS and administered by i.p. injections twice daily at a dose of 1 mg/kg. Control mice received saline i.p.

### 2.4. Experimental Design

In Part A, mice were inoculated with Vĸ*MYC cells (MM mice, *n* = 18) or received a sham injection (*n* = 9) and were followed up with serial echocardiography for 7 weeks (Figure 1A). In Part B, MM mice (*n* = 40) and sham mice (*n* = 46) were followed-up for 6 weeks, and then they were randomized to four subgroups, receiving saline, bortezomib, OAT-1746, or a combination of bortezomib and OAT-1746 for one week (Figure 1B). At the end of the experiment, blood was collected from the heart, the mice were euthanized, the spleen was collected for NO analysis, and the hearts were harvested for Langendorff perfusion and endothelial function studies.

### 2.5. Endothelial Function Studies

The hearts were Langendorff perfused with Krebs–Henseleit buffer. After 20 min of perfusion and coronary flow stabilization, 3 bolus injections of bradykinin (100 pmol) and acetylcholine (100 pmol) were given in 3 min intervals. Each bolus injection induced coronary flow increase lasting for up to 30 s. The effluent from the heart was collected for 30 s after each bolus injection and frozen at −80 °C for nitrite analysis (see below) as a measure of coronary endothelial function.

### 2.6. Nitric Oxide (NO) Measurement in Plasma and Effluent from Isolated Hearts

Blood was collected by cardiac puncture and centrifuged at 2000× *g* for 8 min. Next, the plasma was transferred to a new tube and frozen at −80 °C. Nitrite concentration in the collected samples was determined by a gas phase chemiluminescence reaction of NO with ozone using a Nitric Oxide Analyzer (NOA, Sievers Instruments). In this method, nitrate is reduced to NO gas in the purge vessels of the analyzer by potassium iodide in glacial acetic acid [21].

### 2.7. Echocardiography Imaging

Transthoracic echocardiography was performed using E-cube 15 Platinum (Alpinion Medical Systems, Seoul, Republic of Korea) with a 17 MHz linear transducer with mice being lightly sedated by isoflurane to maintain the heart rate > 400 bpm. After sedation, mice were placed on the heating pad to sustain proper body temperature. Images of the parasternal short-axis view at the papillary muscle level, parasternal long axis view, and the apical 4-chamber view were recorded. LV end-diastolic (LVEDV) and end-systolic (LVESV) volumes were determined from the parasternal long-axis view. LVEF, as a marker of LV systolic function, was calculated from the Simpson method. LV diastolic function was assessed by pulsed Doppler of the mitral inflow, with the use of parameters as E wave to A wave ratio (E/A), isovolumic relaxation time (IVRT), and ejection time (ET). Isovolumic contraction time (IVCT) served as another parameter of contractility. All measurements were obtained by one observer blinded to the study groups.

### 2.8. Statistical Analysis

Data are shown as means ± standard deviation (SD). Sigma Plot 14.0 (Inpixon, Palo Alto, CA, USA) was used for statistical analyses. The normality of data distribution was tested using Shapiro–Wilk test. For statistical analyses of the two groups, an unpaired two-tailed *t*-test was used. For statistical analyses of three or more groups, one-way analysis of variance (ANOVA) was used, followed by posthoc Tukey’s or Dunnett’s multiple comparisons tests. A *p* value of less than 0.05 was considered statistically significant. The survival rate was computed using Kaplan–Meier plots and analyzed with log-rank test. Information on the group size, statistical analysis used, and *p*-values is provided in the figure captions.

## 3. Results

### 3.1. The Model

We used a murine MM model, syngeneic with C57BL/6 mice. Mice were inoculated intravenously with cells isolated from the spleens of diseased transgenic Vκ*MYC mice [19]. There was a progressive increase in the serum monoclonal protein levels starting from week 3 after the inoculation (Figure 2A,I), followed by mortality starting from week 5 (half of the animals were dead by week 7 after inoculation of Vκ*MYC cells, Figure 2B).

### 3.2. Multiple Myeloma Progression Is Associated with Progressive Cardiotoxicity

The parameters of systolic function (ejection fraction [LVEF], a measure of LV systolic function, and isovolumic contraction time [IVCT], a measure of LV contractility) started to decline in week 4 after inoculation of Vκ*MYC cells (Figure 2C and 2D, respectively), coinciding with the increase in monoclonal protein in the plasma (Figure 2A,I), while slightly later LV end-diastolic volume, a marker of LV dilation, started to increase (Figure 2E). Nevertheless, stroke volume, an index of LV performance (Figure 2F), was unchanged. On the other hand, parameters of diastolic function, reflecting the active phase of LV relaxation, LV isovolumic relaxation time [IVRT] (Figure 2G), and passive LV compliance, E/A ratio (Figure 2H), were unchanged throughout the experiment. Representative echocardiography pulsed-wave Doppler image of blood flow through the mitral valve (Figure 2J) and long-axis view LV images (Figure 2K−2N) are shown in Figure 2.

These observations indicate that non-light chain MM in a mouse model results in progressive and selective LV systolic dysfunction and LV dilation while the LV diastolic function is preserved. Moreover, LVEF reached approximately 43% at the time when half of the MM animals were dead (vs. 60–65% in healthy animals), indicating that although this impairment was statistically significant, it was rather mild from the physiological point of view. This is further underscored by preserved overall cardiac performance, indicated by unchanged stroke volume. Therefore, cardiac dysfunction is an unlikely cause of MM animal mortality in our study.

### 3.3. Bortezomib Exacerbates MM-Associated Cardiotoxicity, While an Arginase Inhibitor Prevents It

In the second part of the study, sham inoculated (healthy control) mice and MM mice were followed up for 6 weeks after the inoculation (Figure 1B), and then, at the stage of advanced myeloma, they underwent a week of saline, bortezomib, OAT-1746 or combined (bortezomib+OAT-1746) treatment (see Figure 1B for the experimental design). Neither animal died in the sham inoculated groups. Mortality in the MM mice was unaffected by the provided treatments (7/14, 5/9, 6/13, 5/9 mice died by the end of week 7 in the saline, bortezomib, OAT-1746 and bortezomib+OAT-1746 treatment groups, respectively).

In the healthy mice, bortezomib induced systolic dysfunction, resulting in impairment of LVEF (Figure 3A) and IVCT (Figure 3B), as well as LV dilation (Figure 3C). Stroke volume was preserved (Figure 3D), and again the parameters of diastolic function were unchanged (Figure 3E,F). In MM mice, bortezomib worsened systolic dysfunction induced by MM itself, resulting in a highly significant reduction of LVEF (to 31% vs. 49% in healthy animals treated with bortezomib and 43% in untreated MM animals) and IVCT (Figure 3A and 3B, respectively), without affecting diastolic parameters (Figure 3E,F). While LV dilation did not progress further compared to untreated MM animals (Figure 3C), stroke volume, a parameter of LV performance, was reduced by almost 25%, indicating that the combination of bortezomib and MM induces clinically significant cardiac dysfunction.

An arginase inhibitor completely prevented systolic dysfunction induced by both bortezomib and MM alone but only partially caused by their combination (Figure 3A,B) without affecting diastolic parameters (Figure 3E,F).

### 3.4. Endothelial Function and Nitric Oxide Metabolites in the Plasma and Spleen

To gain insight into the potential mechanism of MM- and bortezomib-induced cardiotoxicity as well as the beneficial effects of an arginase inhibitor, we assessed coronary endothelial function and concentration of NO metabolites in the plasma and spleen.

OAT-1746 improved endothelial function in healthy control mice (but only bradykinin, not acetylcholine response, Figure 4A,B). Surprisingly, OAT-1746 and bortezomib combination improved endothelial function even more.

Untreated MM animals had better endothelial function than untreated healthy animals (Figure 4A,B). Neither bortezomib nor arginase inhibitor affected endothelial function in MM animals, but the combination therapy reduced it compared to untreated MM animals as well as to each of the compounds given as monotherapy (Figure 4A,B).

Plasma concentration of nitrite, a stable metabolite of NO, a measure of systemic NO production, did not differ between untreated healthy and MM mice (Figure 4C). OAT-1746 increased plasma nitrite concentration both in healthy and MM mice but had no effect in either group when given in combination with bortezomib (Figure 4C).

The concentration of nitrite in the spleen homogenate was higher than in plasma only in untreated healthy mice (Figure 4C) and did not differ between specific experimental groups.

## 4. Discussion

Here, we show that (1) multiple myeloma in a mouse model characterized by overproduction of IgG leads to progressive cardiac systolic dysfunction that is rather mild at the time when the animals start to die, suggesting that it is not the primary cause of their death. (2) Bortezomib exacerbates MM-induced systolic dysfunction resulting in significant impairment of cardiac performance. (3) An arginase inhibitor prevents bortezomib- or MM-induced cardiotoxicity but is unable to completely prevent adverse cardiac effects of the combination of MM and bortezomib. (4) Endothelial dysfunction and NO deficiency are not the primary causes of either bortezomib or MM-mediated cardiotoxicity, and improvement of endothelial function is not the primary mechanism of beneficial effects of arginase inhibitors.

### 4.1. Cardiotoxicity Caused by Multiple Myeloma

Here, we report for the first time that non-light chain MM is associated with a progressive isolated LV systolic dysfunction. Although its mechanism remains unknown, we provide several interesting observations: this was obviously not due to increased preload that could indicate high output heart failure reported in humans with MM since stroke volume remained unchanged throughout the MM progression. Coronary endothelial dysfunction and its potential repercussions (myocardial ischemia) also is an unlikely cause since endothelium-mediated NO release was exaggerated in untreated MM animals, as could be expected in view of hyperviscosity related to increased immunoglobulin concentration [10,11]. Moreover, the development of LV systolic dysfunction coincided and correlated with monoclonal protein concentration in our experiments. Possible causes of these adverse cardiovascular effects may include systemic inflammation [22] affecting the heart, renal failure, or hypercalcemia [9].

A fascinating hypothesis is that hyperviscosity itself could impair coronary flow reserve. At rest, maximal resistance to coronary blood flow is offered by the arterioles [23], while during hyperemia, capillaries offer the most resistance, determining the maximal possible increase in coronary blood flow. Hyperviscosity preferentially increases capillary resistance. In one study, tripling of blood viscosity reduced the coronary reserve by 60% [24]. This observation suggests that MM patients with preexisting coronary artery disease warrant special attention since hyperviscosity could result in further impairment of coronary flow at the capillary level, adding to coronary resistance caused by possible atherosclerotic plaques located in large arteries, precipitating ischemia, and its consequences.

### 4.2. Cardiotoxicity of Bortezomib: Summation of Adverse Cardiac Effects of MM and Bortezomib

We have previously shown in healthy rats [16] and mice [12] that bortezomib induced LV systolic dysfunction that was most pronounced after the first doses of the drug and gradually resolved both with continuation of therapy and after its discontinuation [16]. Moreover, it was rather mild and of unlikely clinical significance in a healthy subject. We confirm this observation in the current study and add that bortezomib does not affect LV performance (stroke volume, an index of cardiac efficiency) or diastolic function.

However, when given to MM mice, bortezomib exacerbated LV systolic dysfunction caused by MM itself, resulting in a significant reduction of LV performance. This finding is of particular importance since patients with advanced MM may be at high risk of clinically significant bortezomib-induced cardiotoxicity. Of the proteasome inhibitors used in clinical practice, carfilzomib is the most strongly associated with cardiotoxicity [15], and since cardiovascular adverse effects are the class effect of all PIs, our findings suggest that caution and close cardiovascular monitoring is indicated in MM patients treated with PIs.

We did not investigate the mechanisms of bortezomib cardiotoxicity in this study. However, our previous study in the rat suggests that it affects mitochondrial respiration, targeting complex IV of the electron transport chain [16], eventually causing an energy deficit in the cardiomyocytes. We can hypothesize that this impairment of oxygen utilization could exacerbate problems with oxygen supply caused by MM itself, resulting in an exaggerated energy deficit and contractile abnormalities [25]. Further studies are required to address this possibility.

### 4.3. Endothelial Function and Nitric Oxide Availability in Multiple Myeloma and under Bortezomib Therapy

We used two different compounds, bradykinin and acetylcholine, that act on their respective receptors on the vascular endothelium and stimulate NO production. We used the total amount of NO in the coronary effluent as a measure of coronary endothelial function.

The endothelial function has not been studied yet in untreated multiple myeloma; however, enhanced endothelial responses to bradykinin and acetylcholine found in our study are consistent with the observation that hyperviscosity increases local shear stress, inducing both acute NO release and upregulating endothelial NO synthase. Nevertheless, this improvement did not result in increased systemic NO availability, measured as plasma concentration of stable NO metabolite nitrite. This suggests that improved endothelial function in the MM setting could only indicate the endothelial potential to increase NO production upon stimulation rather than actual baseline NO output. The endothelial function has not been systematically studied in humans with MM. In a single small human study, MM patients had mildly increased plasma NO concentration versus healthy controls, although the variability of the MM patient population reduces the strength of possible conclusions [26]. There is an ongoing clinical trial (NCT03776331) to test vascular endothelial function both in untreated MM and the effects of MM therapy.

Contrary to preliminary human observations [27,28], bortezomib did not affect the endothelial function or plasma nitrite concentrations in our study, in healthy or in MM animals, which makes the endothelial dysfunction an unlikely mediator of adverse cardiac effects of bortezomib in our model.

To verify the hypothesis that large amounts of NO originate from the enlarged spleen of MM mice [29], where inducible nitric oxide synthase could be responsible for NO overproduction, we measured stable nitric oxide metabolite, nitrite in the plasma and spleen homogenates, but nitrite concentration was not increased in either plasma or spleen of MM mice.

### 4.4. An Arginase Inhibitor Protects the Heart from Multiple Myeloma- and Bortezomib-Induced Toxicity

Here, we show that an arginase inhibitor, OAT-1746, protected healthy mice from bortezomib-induced LV systolic dysfunction and LV dilation. Moreover, it was equally effective in preventing MM-induced cardiac systolic impairment; however, its efficacy against adverse cardiac effects of a combination of MM and bortezomib was only partial: while contractility was completely prevented, there was only a trend toward LVEF improvement, while LV dilation was not affected.

Since we have already shown that MM is characterized by increased arginase activity [12] and ʟ-arginine is an essential substrate for NO production [30], we tested if an arginase inhibitor improves NO-mediated endothelial function through preservation of ʟ-arginine supply for NO synthesis. However, OAT-1746 only mildly increased plasma nitrite concentrations in healthy and MM mice, slightly improved endothelial function in healthy mice (only bradykinin response), and even tended to reduce it in MM animals. Surprisingly, the combination of bortezomib and OAT-1746 treatment improved endothelial function in healthy animals but impaired it in MM mice. While the detailed mechanisms behind the effects of an arginase inhibitor on endothelial function and NO production are yet to be understood, it is clear that the beneficial effects of OAT-1746 on cardiac function in our model are not mediated by the endothelium.

Arginase inhibition has already been shown to prevent doxorubicin-induced cardiotoxicity [31], although this effect was mediated mainly by afterload reduction, not by the diminishment of adverse LV remodeling. Improvement of cardiomyocyte calcium handling [32] and reduction of apoptosis and inflammation [33] are other potential mechanisms of beneficial effects of arginase inhibition on cardiac function. We have previously shown that another arginase inhibitor, INCB01158, protected mice from bortezomib-induced cardiotoxicity [12], while sildenafil, a phosphodiesterase-5 inhibitor that reduced degradation of cGMP, an intracellular second messenger of NO, protected bortezomib-treated healthy rats from LV dysfunction [12]. This indicates that the NO pathway, in general, is a promising target for the reduction of MM- and PI-related cardiotoxicity.

### 4.5. Study Limitations

Mouse MM model, despite its similarities with human MM, may nevertheless substantially differ from human disease. We do not perform histological analysis of the hearts; therefore, we cannot exclude the possibility that monoclonal protein deposition or MM cell infiltration was responsible for MM-mediated cardiotoxicity. However, a complete reversal of MM-mediated impairment of LV systolic function by an arginase inhibitor, OAT-1746, given as late as in week 7 when monoclonal protein was highly elevated and there was significant mortality, argues against this possibility.

## 5. Conclusions

In our study, we show that multiple myeloma in a mouse model is characterized by progressive left ventricular systolic dysfunction and that a proteasome inhibitor, bortezomib, exacerbates this dysfunction, while an arginase inhibitor is able to at least partially prevent it. These beneficial or detrimental effects seem not to be mediated by the endothelium.

### Clinical Perspective

Our study suggests that PI should be used with caution, particularly in patients with advanced myeloma, where a summation of cardiotoxic effects could be expected. Therapies aimed at NO pathway, in particular arginase inhibitors, could offer promise in the prevention/treatment of cardiotoxicity in the MM setting. Further studies are needed to address the molecular mechanisms of these effects, but they should not solely focus on endothelial function.

## Figures and Tables

**Figure 1 cancers-15-02191-f001:**
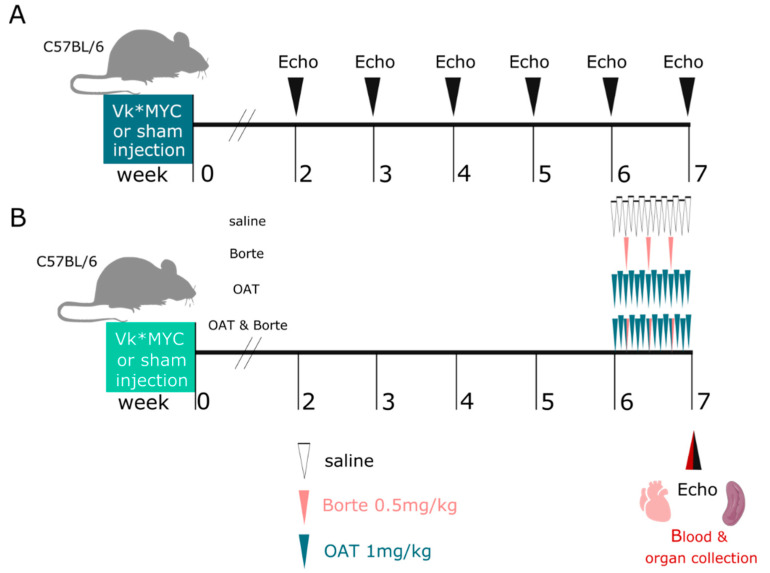
Experimental design In (**A**), mice were inoculated with Vĸ*MYC cells (*n* = 18) or received sham injection (*n* = 9) and were followed up with serial echocardiography for 7 weeks. In (**B**) Vĸ*MYC inoculated mice (*n* = 40) and sham mice (*n* = 46) in week 6 were randomized to four subgroups, receiving saline, bortezomib, OAT-1746, or a combination of bortezomib+OAT-1746 for one week. At the end of the experiment, blood was collected, mice were euthanized, spleens were collected, and the hearts were harvested for endothelial function studies.

**Figure 2 cancers-15-02191-f002:**
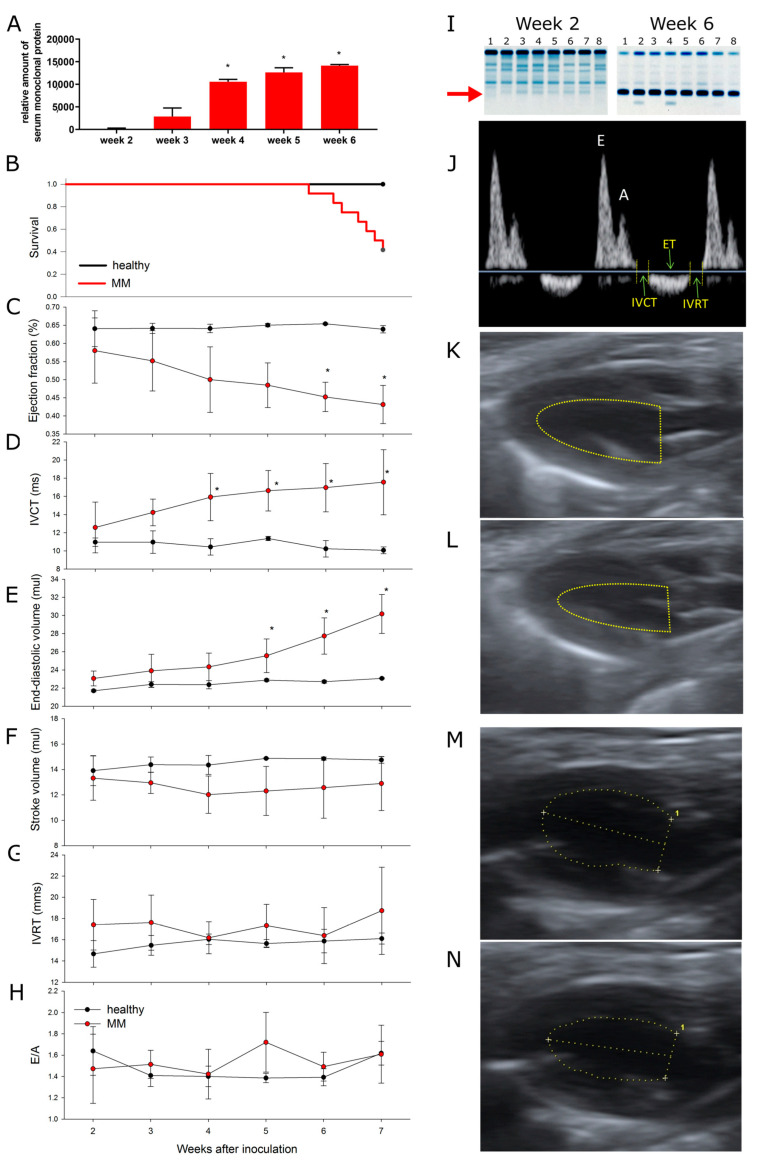
Progression over time of multiple myeloma and cardiotoxicity in a murine model of inoculation with Vĸ*MYC cells. (**A**) Relative amount of serum monoclonal protein, (**B**) Survival of multiple myeloma [MM] and healthy mice after inoculation of MM cells, (**C**) Left ventricular [LV] ejection fraction, (**D**) Isovolumic contraction time [IVCT], (**E**) Left ventricular end-diastolic volume, (**F**) Stroke volume, (**G**) Isovolumic relaxation time [IVRT], (**H**) E wave to A wave ratio [E/A], (**I**) Representative SPEP gels of MM mice serum at week 2 and week 6. While monoclonal protein was undetectable at week 2, the band for monoclonal protein was large in all MM mice at week 6 (red arrow), (**J**) Representative image of pulsed-wave Doppler blood flow through the mitral valve: E wave corresponds to early [E], while A wave to late [A] left atrial filling velocity; E/A ratio is a sensitive measure of passive LV compliance; isovolumic contraction time [IVCT] is a marker of contractility, while isovolumic relaxation time [IVRT] is a marker of active LV relaxation. Representative long-axis view images of LV in a healthy mouse in end-diastole (**K**) and end-systole (**L**) and MM (week 7) mouse in end-diastole (**M**) and end-systole (**N**). While there is no LV dilation and good contractility in a healthy mouse, MM mouse presents LV dilation and poor contractility. Endocardial surface is marked with a yellow dotted line. Means ± SD were shown, *n* = 8 per group. *p* values were calculated with repeated measures ANOVA. * *p* < 0.05. The uncropped blots and molecular weight markers can be found in Appendix A.

**Figure 3 cancers-15-02191-f003:**
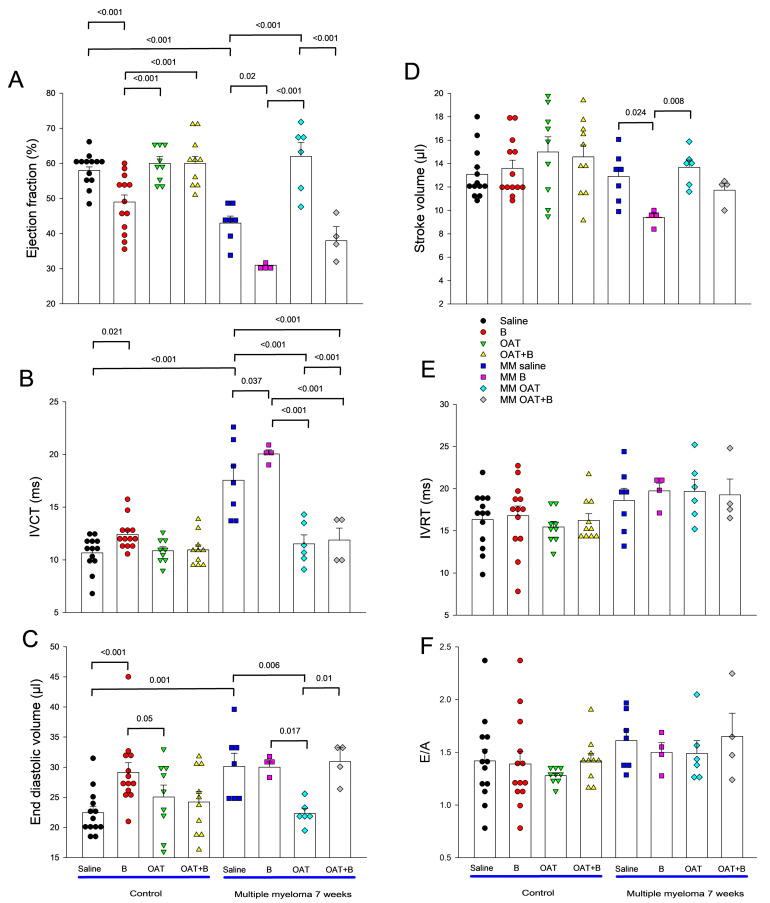
Effects of a proteasome inhibitor, bortezomib, arginase inhibitor, OAT-1746 and their combination on parameters of systolic and diastolic left ventricular function and dilation in healthy and multiple myeloma mice measured by echocardiography. Parameters of left ventricular [LV] systolic function, ejection fraction (**A**) and isovolumic contraction time [IVCT] (**B**), a measure of LV dilation, end-diastolic volume (**C**), a measure of global LV function, stroke volume (**D**), and parameters of LV diastolic function, isovolumic relaxation time [IVRT] (**E**) and E to A wave ratio [E/A] (**F**) and are shown; saline, saline-treated control mice; B, bortezomib-treated control mice; OAT, OAT-1746-treated control mice; OAT+B, OAT-1746 and bortezomib combination treated control mice; MM saline, saline-treated multiple myeloma (MM) mice; MM B, bortezomib-treated MM mice; MM OAT, OAT-1746-treated MM mice; MM OAT+B, OAT-1746, and bortezomib combination treated MM mice; graph presents means ± SD; *n* = 4–14; *p* values were calculated with one-way ANOVA with Tukey’s posthoc test.

**Figure 4 cancers-15-02191-f004:**
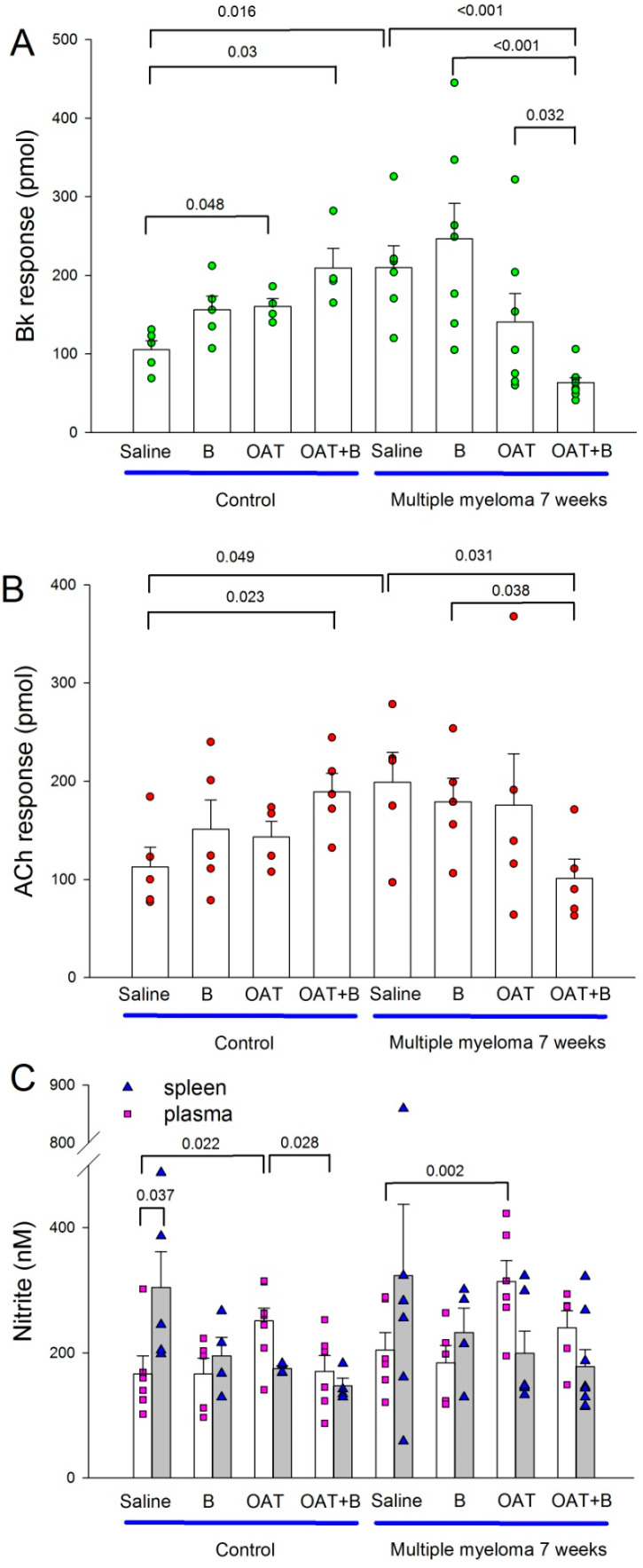
Effects of a proteasome inhibitor, bortezomib, arginase inhibitor, OAT-1746 and their combination on coronary endothelial function as well as plasma and spleen nitrite concentration in healthy and multiple myeloma mice. (**A**,**B**) show coronary endothelial function measured in Langendorff-perfused mouse hearts as an amount of nitrite in coronary effluent over 30 s after bolus injection of bradykinin (Bk response) or acetylcholine (Ach response). Individual data points represent means from 3 measurements. (**C**) presents nitrite concentration in plasma and spleen. Saline, saline-treated mice; B, bortezomib-treated mice; OAT, OAT-1746-treated mice; OAT+B, OAT-1746, and bortezomib combination-treated mice; graph presents means ± SD, *n* = 4–6; *p* values were calculated with one-way ANOVA with Dunnett’s post hoc test.

## Data Availability

Raw data are available upon reasonable request from the corresponding author.

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
