# Peer review of "Arginase Inhibition Mitigates Bortezomib-Exacerbated Cardiotoxicity in Multiple Myeloma"

_cancers, 2023, doi:10.3390/cancers15072191_

Round 1
Reviewer 1 Report
The authors described the effect of arginase inhibition on cardiotoxicity by bortezomib. It seems to be interesting. I have some comments.
1. Is there any data or evidence to prove that arginase-mediated mitigation of cardiotoxicity?
2. I am also interested in the histological evaluation of the heart. If you have data, you should show it.
Author Response
The authors described the effect of arginase inhibition on cardiotoxicity by bortezomib. It seems to be interesting. I have some comments.
- Is there any data or evidence to prove that arginase-mediated mitigation of cardiotoxicity?
Thank you for pointing out this very important issue. Although we do not have any definite answer, in our previous paper (Ramji K et al.,. Scientific Reports. 2022;12:19660) we demonstrated that another, structurally unrelated, orally available, arginase inhibitor, INCB01158 (Incyte and Calithera Biosciences) similarly prevented bortezomib and MM-mediated cardiotoxicity, suggesting, that it is a class effect rather than some unspecific action of OAT-1746 itself. We added this to the Discussion.
- I am also interested in the histological evaluation of the heart. If you have data, you should show it.
Unfortunately we do not have histological data not tissue to perform histological analysis. This is a limitation of the study and we have added this to the “Study limitations” section. We can only speculate that complete reversal of MM-mediated impairment of LV systolic function by an arginase inhibitor, OAT-1746, given as late as in week 7 when monoclonal protein is highly elevated and there is significant mortality, argues against monoclonal protein deposition or cardiac infiltration by MM cells as the primary cause of this cardiotoxicity.
Reviewer 2 Report
In this manuscript by Paterek et al., the authors use the Vkappa myc model of multiple myeloma to investigate cardiac function as a function of time until mice succumb to disease, finding progressively increasing EDV, IVCT upon disease progression. They also use this model to investigate how the PI bortezomib further contributes to cardiac dysfunction in this setting. The are able to show an interesting reversal of cardiac dysfunction with the arginase inhibitor OAT-1746. This manuscript does increase our understanding of PI induced cardiac dysfunction which is an important side effect, particularly of carfilzomib, and offers a new avenue of possible translational relevance with the use of arginase inhibitors in this setting, which are currently in trials. The mechanistic details of this still remain to be clarified.
There are several major critiques that are important to further clarify the interpretation of the results which would further strengthen this study.
1) Was histology performed on the heart of mice as they approach ERC? This would be important to exclude the possibility of either monoclonal deposition or direct myeloma infiltration to ensure that this is not directly contributing to the cardiac dysfunction seen in these mice.
2) Bortezomib given for 1 week causing cardiac dilation in normal mice and it appears that bradykinin, Ach, and Nitrite levels were not statistically different. Despite this, it appears the arginase inhibitor is still normalizing this effect. What do the authors speculate the reason behind this is?
3) Was there any effect of the arginase inhibitor on the growth and progression of the tumor cells themselves? Data on survival and the rise in monoclonal protein should be provided here as this would confound the results of the cardiac studies as this may be more of an effect on the myeloma rather than the endothelium as hypothesized.
Minor issues:
In the results section of the abstract, “endothelial function” or “endothelium” was used, which is quite vague and broad term and can apply to a lot of different possible endothelial processes. Please be much more specific about what is being measured, such as reduced NO levels, increased bradykinin, etc.
Line 26- replace “is not uncommonly” to “can be”
Line 212-213- “Indicating that although this impairment was significant, it was rather mild” Please clarify this sentence as it contradicts itself?
For Figure 4, please provide labels on each x axis throughout the figure
Author Response
In this manuscript by Paterek et al., the authors use the Vkappa myc model of multiple myeloma to investigate cardiac function as a function of time until mice succumb to disease, finding progressively increasing EDV, IVCT upon disease progression. They also use this model to investigate how the PI bortezomib further contributes to cardiac dysfunction in this setting. The are able to show an interesting reversal of cardiac dysfunction with the arginase inhibitor OAT-1746. This manuscript does increase our understanding of PI induced cardiac dysfunction which is an important side effect, particularly of carfilzomib, and offers a new avenue of possible translational relevance with the use of arginase inhibitors in this setting, which are currently in trials. The mechanistic details of this still remain to be clarified.
There are several major critiques that are important to further clarify the interpretation of the results which would further strengthen this study.
1) Was histology performed on the heart of mice as they approach ERC? This would be important to exclude the possibility of either monoclonal deposition or direct myeloma infiltration to ensure that this is not directly contributing to the cardiac dysfunction seen in these mice.
Unfortunately we do not have histological data not tissue to perform histological analysis. This is a limitation of the study and we have added this to the “Study limitations” section. We can only speculate that complete reversal of MM-mediated impairment of LV systolic function by an arginase inhibitor, OAT-1746, given as late as in week 7 when monoclonal protein is highly elevated and there is significant mortality, argues against monoclonal protein deposition or cardiac infiltration by MM cells as the primary cause of this cardiotoxicity.
2) Bortezomib given for 1 week causing cardiac dilation in normal mice and it appears that bradykinin, Ach, and Nitrite levels were not statistically different. Despite this, it appears the arginase inhibitor is still normalizing this effect. What do the authors speculate the reason behind this is?
We observed similar bortezomib-induced cardiotoxicity in the rats (Nowis D et al. Am J Pathol. 2010;176:2658-2668). There we showed that bortezomib affected mitochondrial respiration, targeting complex IV of the electron transport chain, eventually causing energy deficit in the cardiomyocytes. We included this suggestion in the Discussion.
3) Was there any effect of the arginase inhibitor on the growth and progression of the tumor cells themselves? Data on survival and the rise in monoclonal protein should be provided here as this would confound the results of the cardiac studies as this may be more of an effect on the myeloma rather than the endothelium as hypothesized.
Thank you for raising this very important point. Indeed we have previously shown (Ramji K et al.,. Scientific Reports. 2022;12:19660) that in arginase knock-out mice and in WT mice treated with another arginase inhibitor, INCB01158, MM progression was inhibited. However, INCB01158 treatment was provided much earlier in the disease course, in weeks 3 and 4 and for 10 days rather than in week 6 and for 7 days as OAT-1746 was given in this study. Moreover, in our previous study INCB01158 delayed mortality by 2 weeks, while its was unaffected by OAT-1746 in this study (this data was added to section Bortezomib exacerbates MM-associated cardiotoxicity, while an arginase inhibitor prevents it.). This indicates that the arginase inhibitor did not affect the disease progression but rather prevented cardiotoxicity through mechanism not related to MM itself.
Minor issues:
In the results section of the abstract, “endothelial function” or “endothelium” was used, which is quite vague and broad term and can apply to a lot of different possible endothelial processes. Please be much more specific about what is being measured, such as reduced NO levels, increased bradykinin, etc.
Thank you very much for this comment. We have specified that endothelial function refers to “NO production”.
Line 26- replace “is not uncommonly” to “can be”
Corrected.
Line 212-213- “Indicating that although this impairment was significant, it was rather mild” Please clarify this sentence as it contradicts itself?
Thank you for pointing your this contradiction. We have corrected this sentence to: “indicating that although this impairment was statistically significant, it was rather mild from the physiological point of view.”
For Figure 4, please provide labels on each x axis throughout the figure
Thank you for this suggestion. The figure looks much clearer now.
Reviewer 3 Report
This is an elegant and well written preclinical study.
Only some minor comments should be addressed:
-Abbreviations have been used in the abstract. "NO" (line 19) should be also described.
-In the same way, the first time NO is used in the main text (line 63).
-LVEF is described for the first time in line 83, and later again in line 159.
-References should be adapted to the MDPI style.
-The only limitation of the study refers to the mouse MM model. Could you highlight others?
-There is a more updated version of reference 6: Am. J. Hematol. 2022;97:818-829.
Author Response
This is an elegant and well written preclinical study.
Thank you.
Only some minor comments should be addressed:
-Abbreviations have been used in the abstract. "NO" (line 19) should be also described.
Sorry for this, it was corrected.
-In the same way, the first time NO is used in the main text (line 63).
Sorry for this, it was corrected.
-LVEF is described for the first time in line 83, and later again in line 159.
Sorry for this, it was corrected.
-References should be adapted to the MDPI style.
Thank you for pointing this one out. The references were updated.
-The only limitation of the study refers to the mouse MM model. Could you highlight others?
Thank you for pointing this deficiency. We have expanded the Limitations.
-There is a more updated version of reference 6: Am. J. Hematol. 2022;97:818-829.
Thank you for this suggestion. The reference was updated.
Round 2
Reviewer 2 Report
The responses to the previous critiques were well addressed, with references to prior work by the authors that provide greater context for the current study.